# The AIFELL Score as a Predictor of Coronavirus Disease 2019 (COVID-19) Severity and Progression in Hospitalized Patients

**DOI:** 10.3390/diagnostics12030604

**Published:** 2022-02-27

**Authors:** Ian Levenfus, Enrico Ullmann, Katja Petrowski, Jutta Rose, Lars C. Huber, Melina Stüssi-Helbling, Macé M. Schuurmans

**Affiliations:** 1Department of Pulmonology, University Hospital Zurich, 8091 Zurich, Switzerland; mace.schuurmans@usz.ch; 2Faculty of Medicine, University of Zurich, 8032 Zurich, Switzerland; 3Department of Medicine, Technical University Dresden, 01307 Dresden, Germany; enrico.ullmann@uniklinikum-dresden.de; 4Department of Pediatric Psychiatry, Psychotherapy and Psychosomatics, University of Leipzig Medical Center, 04103 Leipzig, Germany; 5Department of Medical Biology, South Ural State University, 454080 Chelyabinsk, Russia; 6Medical Psychology and Sociology, Johannes Gutenberg University Mainz, 55131 Mainz, Germany; kpetrows@uni-mainz.de; 7Clinic for Internal Medicine, Department of Internal Medicine, City Hospital Zurich Triemli, 8063 Zurich, Switzerland; jutta.rose@triemli.zuerich.ch (J.R.); lars.huber@zuerich.ch (L.C.H.); melina.stuessi-helbling@triemli.zuerich.ch (M.S.-H.)

**Keywords:** COVID-19, SARS-CoV-2, score, prediction, severe disease

## Abstract

Since the beginning of the COVID-19 pandemic, SARS-CoV-2 has caused a global burden for health care systems due to high morbidity and mortality rates, leading to caseloads that episodically surpass hospital resources. Due to different disease manifestations, the triage of patients at high risk for a poor outcome continues to be a major challenge for clinicians. The AIFELL score was developed as a simple decision instrument for emergency rooms to distinguish COVID-19 patients in severe disease stages from less severe COVID-19 and non-COVID-19 cases. In the present study, we aimed to evaluate the AIFELL score as a prediction tool for clinical deterioration and disease severity in hospitalized COVID-19 patients. During the second wave of the COVID-19 pandemic in Switzerland, we analyzed consecutively hospitalized patients at the Triemli Hospital Zurich from the end of November 2020 until mid-February 2021. Statistical analyses were performed for group comparisons and to evaluate significance. AIFELL scores of patients developing severe COVID-19 stages IIb and III during hospitalization were significantly higher upon admission compared to those patients not surpassing stages I and IIa. Group comparisons indicated significantly different AIFELL scores between each stage. In conclusion, the AIFELL score at admission was useful to predict the disease severity and progression in hospitalized COVID-19 patients.

## 1. Introduction

Two years after coronavirus disease 2019 (COVID-19) became a worldwide pandemic, the causative agent named severe acute respiratory syndrome coronavirus 2 (SARS-CoV-2) is still a considerable burden for the health care systems of many countries, leading to restrictions of personal movement and business, as well as travel, and sometimes even leading to lockdowns. Despite available vaccines and associated government-funded vaccination campaigns, the spread of SARS-CoV-2 has not been stopped yet and has resulted in recurrent waves of increasing infection rates, which have put a considerable strain on hospitals and, in particular, their intensive care units (ICU). Different mutations of the SARS-CoV-2 spike protein occurred, leading to the emergence of new strains, including the Alpha, Beta, Gamma, Delta and Omicron variants [1,2], which were declared to be variants of concern (VOC) by the World Health Organization (WHO), as they are suspected to cause increased virulence [1,3], higher hospitalization or mortality rates [4,5] or an immune evasion [6,7].

Vaccination campaigns, though initially successful in countries such as Israel [8] or Great Britain [9], have been jeopardized by an incomplete effectiveness of the available vaccines and associated vaccine breakthroughs [7,10], decreasing neutralizing antibody titers over time [11] and vaccine hesitancy in certain parts of the population [12], as well as the limited availability of vaccines in poor countries [13]. For this reason, and due to a rapid spread of the Delta and, later, the Omicron VOC, vaccination campaigns partly suffered setbacks, resulting in further waves of increased infection and hospital occupancy rates [14].

Currently, the main concern is severe cases causing the need for intensive care treatment. Although different testing approaches with sufficient capabilities, including quantitative polymerase chain reaction (qPCR) and rapid antigen tests, are currently widely available in industrialized countries, it is still a challenge to identify patients at high risk for deterioration or severe disease course in emergency rooms, hospital wards or general practice offices in order to give them the necessary attention. Clinical presentations of COVID-19, especially among outpatients, can vary greatly [15] and, identifying and monitoring patients with a high probability of clinical deterioration is crucial for making targeted therapeutic decisions as early as possible.

The AIFELL score was developed and used as a simple frontline triage tool for patients with suspected COVID-19 during the first wave of the pandemic to identify symptomatic patients infected with SARS-CoV-2 from patients presenting with unspecific general or respiratory symptoms in the ER or general practice setting. It includes the patient history components of altered smell or taste and fever, radiologically documented lung infiltrates and the laboratory parameters of C-reactive protein (CRP), elevated lactate dehydrogenase (LDH) and lymphocytopenia (absolute count) [16].

During subsequent increases in infection rates after the first wave, other clinical applications of the simple score were taken into consideration. Since patients in more severe COVID-19 stages II and III (Figure 1) could be identified at hospital admission by the use of this score, the predictive capability of the AIFELL score regarding disease deterioration until hospital discharge was therefore evaluated in hospitalized patients to assess its overall performance as a prediction instrument for COVID-19 progression.

## 2. Materials and Methods

### 2.1. Design

The score was retrospectively applied to consecutively hospitalized SARS-CoV-2-positive patients from 22 November 2020 until 17 February 2021 that were treated at the City Hospital Zurich Triemli (Stadtspital Zürich Triemli) during the second wave of the COVID-19 pandemic in Switzerland. The Department of Internal Medicine of the Triemli Hospital agreed to participate as an external partner to apply the AIFELL score in a different hospital location than where it was initially developed. Patients were either admitted through the emergency rooms (ER) from an outpatient setting or directly to the medical ward or the ICU from other hospitals in the area around Zurich (Switzerland). SARS-CoV-2 swabs for qPCR analyses were either performed in the ER of the Triemli Hospital or by the hospitals and outpatient clinics who referred the patients for further treatment. Inclusion criteria were a positive SARS-CoV-2 swab result analyzed using qPCR at admission, a blood sample available upon admission, including at least two of the three considered laboratory parameters (CRP, LDH and total lymphocyte count), chest imaging (X-ray or computed tomography of the thorax) and documented body temperature. Declaration of consent for the use of health-related data and samples for research purposes was signed by every patient. Exclusion criteria were patients with fewer than two laboratory parameters available and missing chest imaging or documentation of body temperature. In addition, patients were excluded if COVID-19 was not the main reason for hospitalization in order to prevent bias regarding laboratory values. AIFELL scores were calculated for each patient at hospital admission, as described in our previous publication [16]. Due to varying reference ranges for LDH between different hospitals, the original protocol was slightly modified and the LDH limit was defined as the 85th percentile of the maximum value of the LDH reference range, as originally observed in the pilot population at the University Hospital Zurich. Non-measurable LDH results due to hemolysis were also considered to be elevated, which is in line with the fact that hemolysis has been reported to be associated with COVID-19 [17,18]. Since patients with fever often take antipyretic medications before seeing a physician or presenting at the emergency room, attention was paid to history of fever during the last 3 days before hospital admission, and the highest value measured was taken into account. COVID-19 stages according to Siddiqi and Mehra [19] were assigned based on medical records at hospital admission and in the course, especially at disease deterioration. This classification was chosen because the authors suggested specific therapeutic approaches for each stage [19]. Important criteria defining disease stages are described in Figure 1. The initial and the maximum stage during hospitalization were documented to evaluate disease progression. The study was approved by the Institutional Review Board of the Canton of Zurich (Cantonal Ethics Committee, Nr. 2020-03036).

### 2.2. Statistical Analysis

For baseline characteristics, a test for normal distribution and outliers, as well as a z-standardization, were performed on the original raw data. Student’s *t*-test, a multiple linear regression analysis and an ANOVA with multiple measurements were calculated for group comparisons and to determine statistical significance. A multiple linear regression was performed with the AIFELL score at hospital admission to evaluate its predictive capability regarding the maximum COVID-19 stage developed during hospitalization. Next, the course of the illness was analyzed by a two-factorial ANOVA for repeated measurements with within-factor time (pre- and post-measurement) and between-factor group (AIFELL score). In order to account for the influence of sphericity, a Greenhouse–Geisser/Bonferroni correction for repeated testing was implemented. For data collection and statistical analyses, Microsoft Excel 2016 and SPSS 24 were used. Results are given as mean values ± standard deviation (SD) unless indicated otherwise.

## 3. Results

The mean age of the included consecutively hospitalized COVID-19 patients (*n* = 154, 64% male) was 69 years ± 15.7 (range from 23 to 95 years). Baseline demographic, clinical and laboratory characteristics are shown in Table 1.

The average AIFELL score at hospital admission, regardless of the disease stage, was 3.7 ± 0.96. A stage-based comparison of AIFELL scores revealed significant differences between each stage (Table 2). A group comparison of the distribution of AIFELL scores in different disease stages with two-tailed *t*-tests showed highly significant results (*p* < 0.01).

AIFELL scores ≥3 points at admission were associated with development to more severe disease stages (II and III) during the hospital stay, needing a more intensive therapeutic approach (i.e., intermediate or intensive care units). Patients needing oxygen supplementation due to hypoxia during hospitalization (COVID-19 stage IIb and III, *n* = 125) had significantly higher AIFELL scores at admission (3.98 ± 0.75) compared to patients who maximally developed COVID-19 stages I and IIa (2.69 ± 1.11, *p* < 0.0001), as shown in Figure 2.

In those patients who deteriorated during hospitalization (n = 28) from stage IIa/IIb to III or stage IIa to IIb, the average AIFELL score at admission was 3.96 ± 0.83. The average AIFELL score of patients who did not deteriorate during hospitalization (n = 126) was 3.68 ± 0.99, regardless of their disease stage. The mean AIFELL score at admission of patients with mild COVID-19 (stage I), who also did not deteriorate, was 2.08 ± 0.86, and is therefore significantly lower than the score obtained of patients reaching severity stages II and III (*p* < 0.001).

### 3.1. Linear Regression Analysis

In order to predict the course of COVID-19 illness during the hospitalization using the AIFELL score at admission, a multiple linear regression was calculated. This multiple linear regression analysis showed that the AIFELL score at hospital admission significantly predicts the progression of the COVID-19 stage during hospitalization F(1, 152) = 21.459, *p* < 0.001 with R^2^ = 35.2% (Table 3).

### 3.2. ANOVA with Repeated Measures

ANOVA with repeated measures with the Greenhouse–Geisser correction showed a significant time effect of symptom development from admission to the worst stage during hospitalization (F (1, 152) = 8.101, *p* = 0.005). Furthermore, there was a significant group effect (F (1, 152) = 19.867, *p* ≤ 0.001), so the AIFELL score predicted different courses of symptom development (i.e., progression to more severe stages) over time during hospitalization (Table 4). Therefore, the COVID-19 stages I, IIa, IIb and III differ significantly in terms of their AIFELL scores.

## 4. Discussion

In the presented study, the AIFELL score was used as a prediction instrument for disease severity and progression to evaluate its performance in the hospital setting in patients with recently diagnosed COVID-19. Our results indicate that an increased AIFELL score at admission is able to predict the ultimate disease severity or disease progression in patients hospitalized due to COVID-19.

Several scoring systems for COVID-19 with different purposes (triage, prediction) and components have been developed during the pandemic [20,21,22,23,24]. Most of them are considered to be too complex for routine diagnostics since they use advanced formulas for calculation, require extensive prior knowledge regarding medical history and may include a larger number of components or specific laboratory parameters that are not widely or rapidly available in many settings, such as interleukin-2 (IL-2), IL-6, D-dimer, procalcitonin [25] or cluster of differentiation (CD) 4 and CD8 counts, thus limiting their usability in frontline settings or stressful situations. Other scores were developed for specific cohorts (e.g., geriatric patients [26]). Some nonspecific early warning scores have been used in COVID-19 patients in various settings [27,28,29].

The AIFELL score, in contrast, is a simple tool consisting of only basic, low-cost laboratory parameters, patient history and imaging results. It was initially developed based on data from a mixed, representative cohort of COVID-19 patients and studied in consecutive ER patients with suspected COVID-19. It is easy to calculate and does not necessarily require any additional tools, making it practical for resource-limited settings, such as smaller hospitals or general practitioner offices. Nevertheless, the website www.aifell.net was created to provide a comprehensive overview of all components, and features a basic online calculator, as well as a printable calculation form. Due to specified limits and the objective nature of the components, there is less room for inter-rater variability.

Based on the results of this study, we suggest the calculation of the AIFELL score at hospital admission for every patient with respiratory or other symptoms possibly associated with COVID-19, so it may serve as a triage tool and, in the case of a positive SARS-CoV-2 swab, it may additionally serve as a predictor for disease progression. Another reasonable time point for the calculation of the AIFELL score is when disease deterioration has occurred to estimate the probable outcome. Since we demonstrated that the AIFELL score at admission is significantly higher in patients developing more severe COVID-19 stages during hospitalization, it provides useful information for clinical decision making, serving as a red flag for disease deterioration. From a clinical perspective, there is a relevant difference between the COVID-19 stages IIa and IIb, or even III, because hypoxia requires immediate treatment to avoid life-threatening complications. Furthermore, in severe COVID-19, proinflammatory cascades are activated, which eventually may lead to endothelial dysfunction and hypercoagulability, resulting in life-threatening complications, such as myocardial infarction [30] or acute ischemic stroke (AIC), due to inflammation-related plaque rupture [31]. Therefore, patients in stage IIb need hospital care and patients in stage III are usually transferred to the ICU. The AIFELL score could help in identifying these severe cases, since it predicts the expected disease severity. According to our results, patients with an AIFELL score ≥3 points at admission would need more attention and very likely more resources, whereas patients with lower AIFELL scores may require regular ward care or may be discharged due to the expected better prognosis.

Components of the AIFELL score that are associated with severe stages of COVID-19 in the scientific literature are consolidative infiltrates as a sign of pulmonary involvement [32], fever [33] and elevated CRP levels [25,34] as a result of a systemic inflammatory process. Further components associated with severe stages of COVID-19 are elevated levels of LDH, which indicate pronounced tissue damage [35,36] and a lower lymphocyte count [37], possibly due to a chemotaxis of lymphocytes to lymphoid organs or a direct infection and destruction of angiotensin-converting enzyme 2 (ACE2) positive lymphocytes [36].

Dysosmia and dysgeusia tend to be associated with a mild or moderate disease course [38], but play an important role in the triage process for which the AIFELL score was initially generated.

Other inflammation biomarkers, such as the neutrophil count, the neutrophil-to-lymphocyte ratio (NLR) and monocyte-to-lymphocyte ratio (MLR), as well as the platelet count and lymphocyte-to-platelet ratio (LPR), which are usually available or can be calculated after a differential blood count has been performed, could be used to expand or complement the AIFELL score. The importance of detailed analyses of these simple blood parameters has already been highlighted previously [39]. Furthermore, the systemic immune inflammation (SII) index, which is also easily calculated based on lymphocyte, neutrophil and platelet counts and reflects the innate, as well as the adaptive, immune response in COVID-19 and in some other diseases, may potentially add value to the AIFELL score. In particular, the serial calculation of the SII based on repeated laboratory samples of the differential blood count (serial systemic immune inflammation indices, SSIIi) seems to be important, as SSIIi peaks were shown to be associated with neurological or thromboembolic complications [40]. Therefore, SSI and SSIIi may be calculated additionally to complement the information of the AIFELL score.

Biomarkers, such as interleukin-(IL)-1, IL-6, cardiac troponin I, d-dimers, ferritin or procalcitonin, were originally not included as components of the AIFELL score since they are not measured routinely, are usually more expensive and are generally not easily available.

Our study has several limitations: During the second wave, hospitalized patients tended to be more severely ill compared to the first wave. Therefore, less severe COVID-19 stages (I and IIa) are underrepresented in our cohort. It was unknown which strains of SARS-CoV-2 were causing the infection at the time of the study, since sequencing was not routinely performed during the second wave. Other limitations of the study are its retrospective nature and the single-center setting. Due to the retrospective nature of the study, not all data sets were complete. Information about altered smell or taste was missing in many cases, since it was not routinely asked when medical history was obtained. Missing information was rated as negative for the respective component of the score. This may have led to lower average AIFELL scores and explains the low number of patients with the highest AIFELL score of 6 points. Obviously, further research is necessary to validate the AIFELL score as a prediction tool, especially the prospective use of the score in a multicenter setting with a larger cohort of patients. The strengths of the current study are the external application of the score outside of the original hospital setting (University Hospital Zurich) and a mixed cohort of consecutive hospitalized patients during a predefined time frame.

COVID-19 still dominates the medical, social and political discourse, resulting in efforts to develop mechanisms for stopping future waves of the pandemic [41]. During recurring waves with high infection and hospitalization rates, the AIFELL score may continue to be a simple, fast, cost-effective and easily applicable instrument for the triage of outpatients to identify patients prone to developing more severe stages of COVID-19 and to predict their outcome. In particular, low-income countries with fewer resources in their health care systems, and thus a limited availability of advanced laboratory techniques and SARS-CoV-2 testing kits, as well as vaccinations, may benefit from its widespread use [42]. Even in some industrialized countries (e.g., Germany), the rapid spread of the Omicron VOC is pushing PCR testing capabilities to their limits. With large numbers of infected patients, clinical decision instruments to distinguish probable severe cases from rather mild infections are becoming even more important.

Currently available or upcoming IgG1-based monoclonal antibody treatments of COVID-19, such as casirivimab/imdevimab (RegN-Cov2) [43], sotrovimab (Xevudy) [44] or regdanvimab (Regkirona) [45], which can be administered intravenously in an outpatient setting, are intended for patients not yet needing oxygen supplementation but with a high risk of severe disease. As the selection of eligible patients for these therapies may be difficult due to a lack of precise guidelines and disease severity indicators, the AIFELL score could support clinical decision making in these contexts.

In future, after the end of the current COVID-19 pandemic, SARS-CoV-2 may become a relevant respiratory virus of seasonal nature, similar to influenza, with a potential to cause severe complications. Therefore, clinical instruments, such as the AIFELL score, that facilitate triage and enable the prediction of severe stages may remain useful, even after the pandemic situation has largely subsided.

## 5. Conclusions

The AIFELL score is a simple clinical instrument to predict disease severity or disease progression in patients hospitalized due to COVID-19. This finding supplements its capability to be used as a triage tool in the emergency room setting.

## Figures and Tables

**Figure 1 diagnostics-12-00604-f001:**
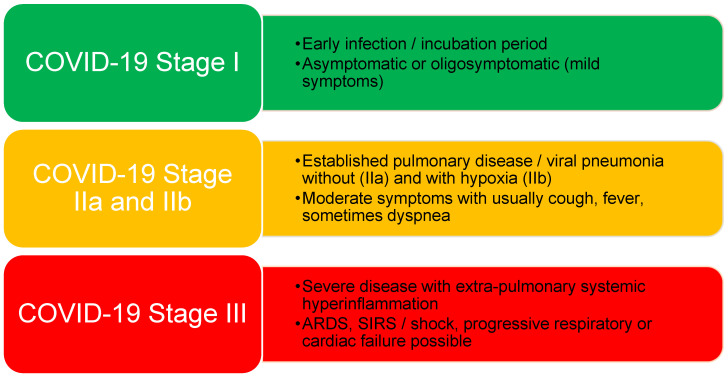
COVID-19 stages as suggested by Siddiqi and Mehra with brief descriptions of their criteria.

**Figure 2 diagnostics-12-00604-f002:**
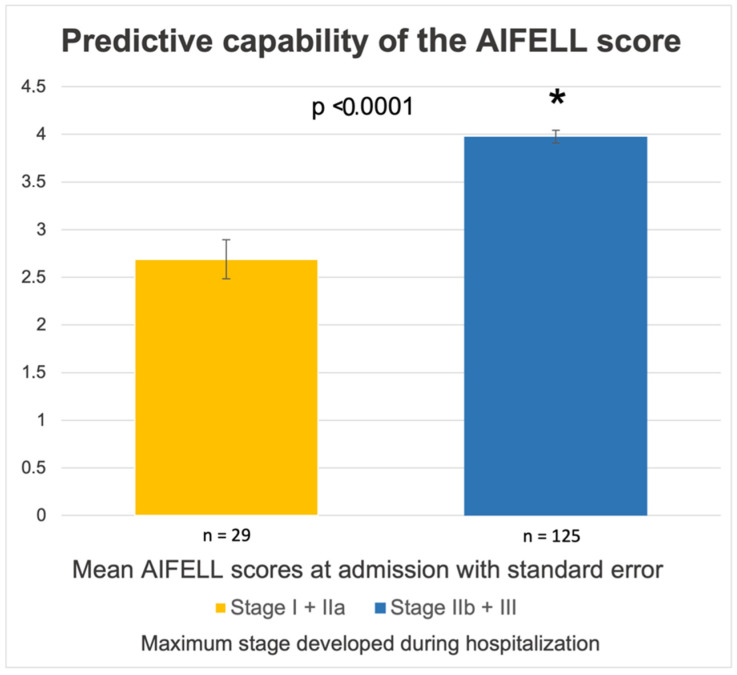
Comparison of mean AIFELL scores at admission between patients who, during the course of the hospitalization, required oxygen supplementation (COVID-19 stage IIb and III) and patients without hypoxia (stage I and IIa). * is a recognized sign for a results being significant.

**Table 1 diagnostics-12-00604-t001:** Baseline characteristics, including COVID-19 stages, AIFELL scores and clinical and laboratory values (*n* = 154).

Baseline Characteristics	Mean (Range)/Total Number (Percentage)	SD
**Demographics**		
Age (years)	69 (23–95)	15.7
Sex (male)	99 (64.2%)	
**Clinical**		
Temperature (°C)	38 (35.6–40)	0.92
SpO2 (%)	92.3 (60–100)	6.75
Heart rate (bpm)	85.7 (57–130)	15.6
Blood pressure (mmHg)		
*Systolic*	135.5 (72–236)	23.5
*Diastolic*	73.7 (40–110)	13.8
Respiratory rate (cycles/Min)	23.6 (8–40)	6.3
Altered smell or taste noted	16 (10.4%)	
Infiltrates documented	142 (92.2%)	
**Laboratory**		
CRP (mg/L)	97 (0.6–422)	79.5
LDH (U/L)	382 (176–1166)	176
Lymphocyte count (10^9^ cells/L)	1.0 (0.1–2.7)	0.465
**Medications administered**		
Antibiotics	70 (45.6%)	
Dexamethason	83 (53.9%)	
Remdesivir	76 (49.4%)	
**Outcomes**		
Ventilation (NIV, intubation)	30 (19.5%)	
Exitus	14 (9.1%)	
**COVID-19 severity stages** **at admission** **according to Siddiqi & Mehra**		
Stage I	13 (8.4%)	
Stage IIa	35 (22.7%)	
Stage IIb	80 (52%)	
Stage III	26 (16.9%)	
**Maximum COVID-19 severity stage during hospitalization**		
Stage I	13 (8.4%)	
Stage IIa	16 (10.4%)	
Stage IIb	87 (56.5%)	
Stage III	38 (24.7%)	
**AIFELL scores at admission**		
0	1 (0.66%)	
1	1 (0.66%)	
2	11 (7.1%)	
3	43 (27.9%)	
4	66 (42.9%)	
5	31 (20.1%)	
6	1 (0.66%)	

NIV, non-invasive ventilation; bpm, beats per minute; CRP, C-reactive protein; LDH, lactate dehydrogenase.

**Table 2 diagnostics-12-00604-t002:** Distribution of the AIFELL scores according to COVID-19 stages.

**Maximum COVID-19 Stage during** **Hospitalization**	**Average AIFELL Score at** **Admission**	**SD**	**Two-Tailed** ***t*-Test**	***p* Value**
Stage I	2.08	0.86	Stage I vs. IIa	0.005
Stage IIa	3.2	0.05	Stage I vs. II & III	<0.0001
Stage IIb	3.9	0.77	Stage IIa vs. IIb	0.002
Stage III	4.2	0.65	Stage II vs. III	0.008
TOTAL	3.73	0.96		

**Table 3 diagnostics-12-00604-t003:** Multiple linear regression of COVID-19 maximal stage during hospitalization by AIFELL score.

	*b*	SE_B_	*β*	*T*	*p*
AIFELL score	0.677	0.146	0.352	4.632	0.000 ***

Note. R^2^ = 0.352. *** *p* < 0.001.

**Table 4 diagnostics-12-00604-t004:** Results of ANOVA with repeated measures comparing AIFELL scores at admission and worst disease stage reached during hospitalization.

Parameter	ANOVA Time Effect		ANOVA Group Effect		ANOVA Time * Group Interaction	
	*F*	*P*	*F*	*p*	*F*	*p*
AIFELL score	8.101	0.005 **	19.867	0.000 ***	1.276	0.26

* *p* < 0.05. ** *p* < 0.01. *** *p* < 0.001.

## Data Availability

Not applicable.

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
