# Peer review of "The AIFELL Score as a Predictor of Coronavirus Disease 2019 (COVID-19) Severity and Progression in Hospitalized Patients"

_diagnostics, 2022, doi:10.3390/diagnostics12030604_

Round 1
Reviewer 1 Report
Overall the paper is well written. Expand the discussion the utility of easily available biomarekers beyond what is described here, at least in the discussion.
1.https://www.jns-journal.com/article/S0022-510X(21)00049-6/fulltext
The simple caculation of serial systemic immune inflmmatory indices appear to add value. You may suggest expanding the score using such scores that can easily be automated globally.
Suggest expanding on the pathophysiology and elaborate on expanding beyond what authros have tested here as the mansucript as it is not making the reader wiser than earlier.
See below
https://www.frontiersin.org/articles/10.3389/fneur.2020.607221/full
Please revise and improve this work further.
Author Response
Point-by-point response to the comments of the reviewers:
Reviewer 1
“Overall the paper is well written. Expand the discussion, the utility of easily available biomarkers beyond what is described here, at least in the discussion.
Response: We thank you for the comment and the valuable proposal. As suggested, we have expanded the discussion section of the manuscript to include the easily available biomarkers that have been mentioned in the two publications listed in your reviewer comment.
“https://www.jns-journal.com/article/S0022-510X(21)00049-6/fulltext
Those easily available biomarkers [should be mentioned here].
The simple calculation of serial systemic immune inflammatory indices appear to add value. You may suggest expanding the score using such scores that can easily be automated globally.”
Response: We agree that the systemic immune inflammatory index (SII) and its serial calculation may add value to the AIFELL score. Therefore, we added additional passages regarding SII and SSIIi and cited the publication you mentioned.
“Suggest expanding on the pathophysiology and elaborate on expanding beyond what authors have tested here as the manuscript as it is not making the reader wiser than earlier.
See below
https://www.frontiersin.org/articles/10.3389/fneur.2020.607221/full
Please revise and improve this work further.”
Response: Thank you for your suggestion. Based on you comment we expanded the ‘Discussion’ section and summarized the pathophysiology of the included laboratory components in the AIFELL score.
We hope that the clarified presentation of our results by providing additional information as suggested by the reviewers has explained the raised questions.
We would like to thank the reviewers for their questions, comments and suggestions.
The manuscript has benefitted strongly from the review process and we hope it is now acceptable for publication.
Ian Levenfus & Macé Schuurmans for the author team
Reviewer 2 Report
1. Your main finding is that AIFELL scores ≥ 3 points at admission were associated with more severe disease stages (II and III) during hospital stay, needing a more intensive therapeutic approach (i.e. intermediate or intensive care units). How is this different from your original study in the ER?
2. Results: In those patients who deteriorated during hospitalization (n = 27) from Stage IIa/IIb to III or Stage IIa to IIb, the average AIFELL score at admission was 3.93 ± 0.83.
What about those who did not deteriorate. What was their score?
3. Linear regression
A multiple linear regression analysis showed that the AIFELL score calculated at hospital admission significantly predicts the COVID-19 stage developed during hospitalization.
What do you mean "predicts the COVID-19 stage developed during hospitaliztion"? You have no data/results to support this. It merely reflects the status of the patient at that time.
4. Results: AIFELL scores of patients developing severe COVID-19 Stages IIb and III during hospitalization were significantly higher upon admission compared to Stages I and IIa.
But you don't tell us the initial stage of the patient.
The results need to be clarified. Missing Table with patient characteristics (scores, COVID-19 stage, demographics, etc etc).
More analyses are required in order to support conclusions.
Author Response
Point-by-point response to the comments of the reviewers:
Reviewer 2
“1. Your main finding is that AIFELL scores ≥ 3 points at admission were associated with more severe disease stages (II and III) during hospital stay, needing a more intensive therapeutic approach (i.e. intermediate or intensive care units). How is this different from your original study in the ER?”
Response: In contrast to our initial study in the ER, the setting as well as the patient population is different. In the 1st study, consecutive ER patients with or without COVID-19 showing symptoms which may be suggestive for COVID-19 diagnosis were evaluated. The current 2nd study deals with consecutively hospitalized patients with PCR-proven COVID-19.
During the 1st study, the COVID-19 swab result as well as the disease stage were evaluated at the timepoint of ER presentation prior to hospitalization. The aim then was to determine whether it is possible to select probable COVID-19 cases for hospitalization in spontaneously presenting or referred patients with acute respiratory symptoms. The current 2nd study, on the other hand, evaluated the disease development/progression from the initial disease stage at hospitalization to the maximum (worst) developed disease stage/severity during hospitalization and whether a prediction of disease progression during hospitalization is possible based on the AIFELL score at admission. Additionally, the study was performed at a different hospital. These aspects have more clearly been described and discussed.
“2. Results: In those patients who deteriorated during hospitalization (n = 27) from Stage IIa/IIb to III or Stage IIa to IIb, the average AIFELL score at admission was 3.93 ± 0.83.
What about those who did not deteriorate. What was their score?”
Response: Thank you very much for pointing out the missing score for those that did not deteriorate. We have added this information in the ‘Results’ section (lines 179 – 185). An additional correction had to be made: The total number of patients who deteriorated during hospitalization was n = 28, not 27. The mean value of patients who deteriorated was therefore recalculated and changed slightly (now 3.96, before 3.93). This did not affect the overall conclusions in any way.
“3. Linear regression
A multiple linear regression analysis showed that the AIFELL score calculated at hospital admission significantly predicts the COVID-19 stage developed during hospitalization.
What do you mean "predicts the COVID-19 stage developed during hospitalization"? You have no data/results to support this. It merely reflects the status of the patient at that time.”
Response: Thank you very much for this comment! We revised our explanation of the statistical analyses in the ‘Results’ section. In this study we specified the COVID-19 disease stage twice (at different time points) during hospitalization and therefore captured the course of the COVID-19 illness over time. These developments were predicted in a multiple linear regression model based on the initial AIFELL score at admission. This means that the elevated AIFELL score acts as a red flag in certain patients, who have not reached maximal disease severity at the time when the score is calculated, but the fact that the calculated score is high (≥3 points) suggests/predicts that the condition is likely to worsen within a few days. This means that a higher AIFELL score in a patient with a stable non-severe COVID-19 condition (Stage I or IIa) serves as predictor for an expected progression to a more severe stage within days. This has been explained better in the manuscript. Additionally, we revised the description of the multiple linear regression in the subsection ‘Materials and Methods/Statistical analysis’.
“4. Results: AIFELL scores of patients developing severe COVID-19 Stages IIb and III during hospitalization were significantly higher upon admission compared to Stages I and IIa.
But you don't tell us the initial stage of the patient.
The results need to be clarified. Missing Table with patient characteristics (scores, COVID-19 stage, demographics, etc.).”
Response: Thank you for pointing out this omission, which is a very important shortcoming of the first version of the manuscript. We revised the description of the results and added the Table 1 in which the demographics, the initial stages as well as the AIFELL scores and their percentages are presented.
We hope that the clarified presentation of our results by providing additional information as suggested by the reviewers has explained the raised questions. Furthermore, we believe that the conclusion is now more clearly supported by data.
We would like to thank the reviewers for their questions, comments and suggestions.
The manuscript has benefitted strongly from the review process and we hope it is now acceptable for publication.
Ian Levenfus & Macé Schuurmans for the author team
Round 2
Reviewer 2 Report
I thank the authors for clarifying and correcting the ambiguous parts of their manuscript.